# Explaining predictive factors in patient pathways using autoencoders

**Hugo De Oliveira**[1,2]*, **Prodel Martin**[2], **Lamarsalle Ludovic**[2], **Augusto Vincent**[1], **Xie Xiaolan**[1]

**1** Mines Saint-Étienne, Univ Clermont Auvergne, CNRS, UMR 6158 LIMOS, Centre CIS, Saint-Étienne, France, **2** HEVA, Lyon, France

* de_oliveira_h@outlook.com

## Abstract

This paper introduces an end-to-end methodology to predict a pathway-related outcome and identifying predictive factors using autoencoders. A formal description of autoencoders for explainable binary predictions is presented, along with two objective functions that allows for filtering and inverting negative examples during training. A methodology to model and transform complex medical event logs is also proposed, which keeps the pathway information in terms of events and time, as well as the hierarchy information carried in medical codes. A case study is presented, in which the short-term mortality after the implementation of an Implantable Cardioverter-Defibrillator is predicted. Proposed methodologies have been tested and compared to other predictive methods, both explainable and not explainable. Results show the competitiveness of the method in terms of performances, particularly the use of a Variational Auto Encoder with an inverse objective function. Finally, the explainability of the method has been demonstrated, allowing for the identification of interesting predictive factors validated using relative risks.

**Data Availability Statement:** Data cannot be shared publicly because the data contain potentially identifying or sensitive patient information. Data have been provided by the Health Data Hub and the French National Commission on Informatics and

## 1 Introduction

Electronic Health Record (EHR) systems have been firstly created to improve care delivery. Their number strongly increased in the past few years, and so did the secondary use of these databases. Among EHR databases, non-clinical claims databases are promising but challenging. The French national health insurance database (SNIIRAM) is one of these. It contains healthcare reimbursements of almost all French citizens since 2006 [1]. The amount of data is colossal, with 98.8% of the 66 million inhabitants French population in 2016 [2]. The main interest of this database is its exhaustiveness as all patients' hospitalizations, medical visits and drug prescriptions are recorded. Despite the inherent complexity [3] of this database (an extensive number of tables, centered on reimbursement and with complex relations) and even if precise medical information is not recorded (such as test results, imaging reports, or vital signs) the SNIIRAM is resourceful, particularly for pharmacoepidemiology studies [2].

This paper focuses on the problem of predicting a pathway-related outcome and identifying predictive factors. Examples of such binary classification tasks in healthcare include the

Liberty (https://www.cnil.fr/en/contact-us, authorization number DR-2019-122) after positive evaluation by the Expertise Committee for Research, Studies and Evaluations in the field of Health (https://www.health-data-hub.fr/cesrees, project number TPS 347167).

**Funding:** The author(s) received no specific funding for this work.

**Competing interests:** The authors have declared that no competing interests exist.

prediction of relapse, the occurrence of a surgery, and the mortality within a period of time. This problem has been largely tackled in the literature, where the number of case studies involving deep learning methods skyrocketed [4].

However, some challenges remains, particularly regarding *trust*: (i) trust in the administrations which use personal health data, but also (ii) trust in the algorithms when it comes to predictions. For the first aspect, the use of data with a very low risk of patient identification is valuable. A particularity of the present study is to only focus on pathways data, without any other patient-centered information such age, gender, ethnicity or localization. As far as the second aspect of trust is concerned, the production of interpretive predictions is a key challenge for actual and future work [4–6]. Naturally, quantitative performances are a necessary condition for the validation of deep learning-based predictive tools. But the explanation of the predictions is essential (i) to simplify the practical deployment of such a novelty at a national level, (ii) to help the comprehension of hidden patterns discovered by a model, and (iii) to enable knowledge discovery regarding patient pathways. Such promising discoveries could be the causalities between a medical event and a selected outcome, the early detection of drug side effects, or the highlight of compliance failures.

Another challenge is the modeling of raw data describing patient pathways and medical events in such databases. Medical events are often completed using standard medical codes. These codes give inputs regarding events such as hospitalizations or medical visits, with precise information related to diagnostics, medical procedures, devices or drugs. They are taken from widely used classification systems, such as the International Classification of Disease (ICD) for diagnoses or the Anatomical Therapeutic Chemical (ATC) classification system for drugs, and organized in a hierarchical structure, with different levels of aggregation. In practice, the selection of the right aggregation level depends on the pathology studied and often necessitate experts' input.

These medical events, ordered in time, constitutes an event log of the patient pathway. Here, the modeling of time is crucial in predictive analytics. If a given medical event may influence future outcomes, the time between these events may influence the prediction (for example long- or short-term before inclusion). The repetition of an event during the medical history may also be impacting (for example a single event or the multiple repetition of such event in time).

As a result, this paper focuses on time and codes when modeling medical event logs, and introduce an end-to-end methodology to predict pathway-related outcomes, while identify predictive factors from raw data using autoencoders. Motivated by recent works where autoencoders were used to automatically label complex medical event logs in an explainable way [7], this paper extends these ideas and investigates their use in a predictive setting, with the consideration of time. The main contributions of this paper are (i) a formalization of the use of autoencoders for explainable predictions; and (ii) a methodology to model and transform complex medical event logs, keeping the pathway information. These contributions focus on:

1. modeling time in pathway sequences to make the extraction of relevant patterns possible;

2. dealing with the hierarchical structure of codes;

3. providing a visual explanation of what has been learned by the model;

4. validating the predictive factors extracted through relative risks, widely used in bio-statistical analysis.

The rest of this paper is organized as follows. Section 2 presents related works. A formal description of autoencoders for explainable predictions is introduced in Section 3. The

methodology to model and transform complex medical event logs is presented in Section 4. To validate the presented contributions, a case study using the SNIIRAM database is presented in Section 5. Finally, Section 6 expose the conclusions.

## 2 Related work

EHR data are valuable resources to understand the natural history of disease, quantify the effect of an intervention, construct evidence-based guidelines or detect adverse events [8]. A consistent part of these recent studies relies on deep learning methods, which switch from expert-defined to data-driven feature creation [6]. Doctor AI [9] has been presented in 2016 to perform differential diagnosis from EHR data. Miotto et al. presented Deep Patient [5] in 2016, an unsupervised method to represent patients from EHR. To predict the probability of disease appearance, a random forest algorithm was trained over the representations, giving better performances than the original representation or other dimension reduction methods. A global study focused on scalability was performed by Rajkomar et al. [10], where various targets and models were used. In order to perform patient clustering, Landi et al. [11] used a convolutional autoencoder to learn a latent representation of patients.

In the literature, few studies use machine learning methods for the analysis of French non-clinical claims data. Focused on hospital data, a benchmark of classic machine learning methods has been presented in 2018 [12]. Using similar methods, unplanned 30-day rehospitalizations have been predicted by Jaotombo et al. in 2020 [13]. Also, Cavailles et al. used a machine learning model in order to identify high risk patient profiles within the population [14]. Regarding the SNIIRAM database, Janssoone et al. [15] compared multiple models to predict medication non-adherence using this database. Kabeshova et al. presented ZiMM ED [16], a predictive model for the long-term prediction of adverse events. To the best of our knowledge, no other paper presented prediction methods on this database.

Explainable artificial intelligence (or XAI) is an active topic of research, and many recent studies on EHR focuses on that subject, as shown by Payrovnaziri et al. in their recent systematic review [17]. Among the five methods to explain predictions identified by the authors of this review, the use of attention mechanisms for deep sequential models has been highlighted. RETAIN, introduced by Choi et al. [18], is one of them. Interpretations are provided for a given patient, by giving the importance of each element in its history.

More generally, some model-agnostic explainable frameworks have been introduced to explain black-box models. LIME [19] and SHAP [20] are examples of such frameworks. The first one uses linear models to approximate local behaviours. The second one uses Shapley values for both global and local interpretability. A large number of recent studies involving EHR data and focused on explaining predictions of black-box models use SHAP [21–26]. For example, XGBoost has been explained when predicting the recurrence of breast cancer [21], the early detection of sepsis [22], and the occurence of acute myocardial infarction [23]. Wong et al. [24] also explained the predictions of LightGBM for the unplanned 30-day readmission of patients with cancer. Regarding deep neural networks, SHAP appears useful to explain prediction of 1-dimensional convolutional networks, in order to compute early warning score for the early detection of acute critical illness [25]. LSTM has also been explained in the prediction of 90-day mortality in intensive care units [26].

The common use of SHAP in the literature to explain black-box models is motivated by its solid theoretical foundation, a well documented and easy-to-use framework, as well as a fast implementation to explain tree-based black-box models [27]. One limitation of the use of model agnostic frameworks is the difficulty of correctly interpreting the explanations, without leading the user to draw false deductions without considering the limitations of the model

itself, and obscure the complexity of the study treated. Moreover, a recent comment by Rudin [28] in 2019 arbitrates for the use of intrinsically interpretative models for high stakes instead of trying to explain black box models.

Representation learning [29] draw intention of many research fields, particularly in healthcare [30]. The automatic creation of an adapted representation of medical concepts have been investigated in the past few years. Med2Vec [31], GRAM [32], and more recently Cui2vec [33] are notable examples of the representation of medical concepts in EHR data. However, these embedding vectors are not easily interpretable for medical stakeholders.

The explanation of temporal patterns also remains an interesting research track. The representation introduced by Wang et al. [34] in 2013 is one example. The two-dimensional representation proposed has been successfully used to mine signatures from patient pathways. Another support for temporal visualization of event logs is process mining. As an example, Prodel et al. [35] proposed an algorithm for raw event logs processing. Applied on a case study using patient pathways, a particular focus on integrating the hierarchy of codes during the optimization process has been presented. In order to properly model time, an improvement of the previously mentioned work has been recently proposed [36], with an extension for prediction [37].

As a result, most recent studies used complex embedding and black box models to process complex health data. These methods have been accurate and successful in predictive tasks. The challenge of explainability for such models, and more generally for black-box models, has been an active research topic recently. However, the consideration of complex medical event logs in that context still remains an open challenge. And even if the consideration of time has been treated in the literature, providing a formalism adapted to predictive modeling, explanation and graphical representation is still an open topic for research. Moreover, widely used medical codes and their complex hierarchical structures may be difficult to process, particularly in the context of explainability. Finally, proposing a methodology to validate explainable factors through widely used bio-statistical metrics seems valuable to improve trust in results and facilitate the collaboration with medical stakeholders. Thus, to the best of our knowledge, no end-to-end modeling of both time and hierarchical medical codes that allows for prediction with global explainability and validation of predictive factors has been proposed in the literature.

## 3 Autoencoders for explaining predictive factors

### 3.1 Motivations

The objective of this paper is to introduce a methodology that can predict pathway-related outcomes, while identifying predictive factors from raw data. The core of the proposed method relies on autoencoders, a method developed by the field of representation learning to learn a lower dimensional representation of data [29]. The objective of this section is to provide the necessary formalism to introduce the proposed method, used to predict and explain predictive factors of short-term mortality after the implementation of an Implantable Cardioverter-Defibrillator.

### 3.2 Autoencoder formalism

Let $x = (x_{i,j})_{i \in [\![1,l]\!], j \in [\![1,w]\!]}$ be a 2-dimensional matrix, such as $\forall i \in [\![1, l]\!], \forall j \in [\![1, w]\!], x_{i,j} \in [0, 1]$. The total number of parameter of $x$ is denoted $p = l \times w$. Let $X = \{x^k\}_{k \in [\![1,n]\!]}$ be the set of all elements $x^k$ of size $n$. Autoencoders consists in two parts: an encoder $f_\theta$ and a decoder $g_\theta$. The encoder takes input data $x$ and returns a lower dimensional representation $z = f_\theta(x)$, the low dimensional space of $z$ being referred to as the latent space. The decoder $g_\theta$ maps $z$ back to the high-dimensional initial space $g_\theta(z) = x'$. Here, $f_\theta$ and $g_\theta$ are deep neural networks,

parametrized by their respective weights and bias $\theta$. In order to train an autoencoder, the general idea is to minimize the following objective function:

$$\mathcal{J}_\theta = \sum_{x \in X} L(x, x') = \sum_{x \in X} L(x, g_\theta(f_\theta(x))) \tag{1}$$

where $L(x, x')$ compute the cross entropy between $x$ and $x'$. Because of the low dimensionality of the latent space, this bottleneck architecture constrains the autoencoder to learn only data properties which are useful for reconstruction.

In order to preform binary classification through autoencoders, different strategies can be used. In this paper, a strategy is defined by (i) a loss function $\mathcal{J}_\theta$; (ii) a predictive method with the definition of the corresponding decoding output $y$; and (iii) an explanation element $\mathcal{E}$ that summarises the output predictive factors identified. In the following, the elements defining strategies are formalized, around two strategies referred to as "filter" and "inverse".

### 3.3 Objective functions

In the context of binary classification, the class function $C(x) \in \{0, 1\}$ returns the corresponding class of $x$. The filter objective function $\mathcal{J}_\theta^F$ and the inverse objective function $\mathcal{J}_\theta^I$ are defined such as:

$$\mathcal{J}_\theta^\alpha = \sum_{x \in X} L(\delta^\alpha(x), x') \tag{2}$$

with $\alpha \in \{F, I\}$ for "filter" and "inverse", respectively. The functions $\delta^F(x)$ and $\delta^I(x)$ are defined such as:

$$\delta^F(x) = \begin{cases} 0_x & \text{if } C(x) = 0 \\ x & \text{if } C(x) = 1 \end{cases} \tag{3}$$

$$\delta^I(x) = \begin{cases} 1_x - x & \text{if } C(x) = 0 \\ x & \text{if } C(x) = 1 \end{cases} \tag{4}$$

with $0_x$ and $1_x$ defined as elements with the shape of $x$ but full of 0 or 1 values, respectively.

By minimizing $\mathcal{J}_\theta^F$ during the training, the autoencoder learns to reconstruct positive elements while reconstructing zeros for the negative ones. Once trained, the autoencoder works as a filter, which only lets the information of positive element being decoded. By minimizing $\mathcal{J}_\theta^I$, the autoencoder learns to reconstruct positive elements while reconstructing inverse of the input for the negative ones, working as a classifier but with multidimensional output.

### 3.4 Predictive methods

Following the idea behind the construction of the previously defined loss functions, the higher the amount of decoded information, the higher the probability of an element $x$ to be positive. Then, the output $y^F(x)$ is used for the prediction by measuring the amount of decoded information of $x$, and is defined as:

$$y^F(x) = \sum_{\substack{i \in [\![1,l]\!] \\ j \in [\![1,w]\!]}} x'_{i,j} \tag{5}$$

An autoencoder trained by $\mathcal{J}_\theta^I$ learns to detect patterns of positive class in order to return an

inverse of the input for negative class. As a result, the lower the reconstruction error between $x$ and $x'$, the higher the probability of $x$ to be positive. Thus, the inverse error of reconstruction $y^I(x)$ is used to predict and is defined as:

$$y^I(x) = 1 - \frac{1}{p}\sqrt{\sum_{\substack{i\in[\![1,l]\!]\\ j\in[\![1,w]\!]}} |x_{i,j} - x'_{i,j}|^2} \tag{6}$$

### 3.5 Explanation element

Once trained using objective function (2), the autoencoder can reveal a global explanation of the predictions. The negative set $X_0$ and positive set $X_1$ are defined such as $X_c = \{x \in X | C(x) = c\}$ for $c \in \{0, 1\}$. The number of elements in $X_c$ is noted $n_c$. The mean element of class $c$ is noted $\bar{x}_c$ and is defined as

$$\bar{x}_c = \frac{1}{n_c}\sum_{x\in X_c} g_\theta(f_\theta(x)) \tag{7}$$

Mean elements of positive and negative classes are then used to compute a global explanation of what was learned by the autoencoder to classify. Thus, the explanation element noted $\mathcal{E}$ is introduced and defined as:

$$\mathcal{E} = \bar{x}_1 - \bar{x}_0 \tag{8}$$

The explanation element is then a matrix of same dimensions as the input elements, which summarizes all the predictive factors that have been identified during the training. It is defined by the difference between mean decoding of positive and negative elements. Because both objective functions $\mathcal{J}_\theta^F$ and $\mathcal{J}_\theta^I$ tends to extract patterns of the positive class, $\mathcal{E}$ highlight characteristic elements of $x$ which positively impact prediction.

A summary of all notations is proposed in Table 1. In the next section, details regarding the structure of $x$ in the context of patient pathways are provided.

## 4 Complex medical event logs modeling

The modeling of complex event logs is challenging, particularly for explainable predictive tasks. In this section, a general methodology is presented to tackle these challenges, illustrated

**Table 1. Notations summary.**

| Name | Notation |
|---|---|
| Matrix | $(x_{i,j})_{i\in[\![1,l]\!], j\in[\![1,w]\!]}$ |
| Dimension of $x$ | $p = l \times w$ |
| Set of $x$ | $X = \{x^k\}_{k\in[\![1,n]\!]}$ |
| Set of class $c$ | $X_c$ |
| Encoding function | $f_\theta$ |
| Decoding function | $g_\theta$ |
| Filter objective function | $\mathcal{J}_\theta^F$ |
| Inverse objective function | $\mathcal{J}_\theta^I$ |
| Filtered output | $y^F(x)$ |
| Inverse output (error of reconstruction) | $y^I(x)$ |
| Mean element of class $c$ | $\bar{x}_c$ |
| Explanation element | $\mathcal{E}$ |

through a complete example presented in Fig 1. Even if focused on patient pathways, this method can be applied to other types of event logs and contribute to other fields.

In the context of event logs representing patient pathways, each medical event is represented by a set of medical activities, each activity having a given label which corresponds to a medical code from classification systems. These classification systems are often organized in hierarchical structures. As a result, each label is inherently completed by its hierarchical knowledge. Some modeling choices are highly dependent of the considered case study. This is the case for the choice of the labels used to describe medical activities. One the one hand, the use of medical codes is the most precise choice (its label as found in the system, at the lowest level in the hierarchy). On the other hand, selecting a higher level in the hierarchy can bring more medical insight and improve interpretability. In order to not introduce user bias in the modeling, all levels of the hierarchy are kept in the proposed modeling. As a result, for each activity of a given event, activities with labels corresponding to higher hierarchy levels are added to the event log. This results in a larger event log, enriched with the hierarchical information.

The resulting event log gather the entire input information. However, this representation is not suitable for user interpretation and explainability. Therefore, the information of each patient for the event log will be represented as a 2-dimensional matrix $x$ where each row represents an activity and each column is a time window. This representation is similar to the *Temporal Event Matrix Representation (TEMR)* introduced by Wang et al. [34], with the notion of time windows to modify the time scale for long-time follow-up studies. The total number of different labels representing activities are noted $l$ and the total number of time windows $w$. Thus, for a patient $x = \{x_{i,j}\}$, for $i \in [\![1, l]\!]$ and $j \in [\![1, w]\!]$, $x_{i,j}$ is the number of occurrence of activity $i$ during the time window $j$. Example 1 illustrated both the data modeling methodology and the predictive method using a synthetic event log.

**Example 1** *After transformation, each synthetic patient of the event log is represented by a matrix, with $l = 7$ activities and $w = 26$ time windows. Once trained using $\mathcal{J}_\theta^F$ as objective function and 80% of the event log, the autoencoder is able to reconstruct positive elements while reconstructing zeros for the negative ones. Fig 2 presents an example of negative (a) and positive (b) elements from the remaining 20% of the event log, before (left) and after (right) being encoded and decoded by the trained autoencoder. The results show that the information from the negative element is filtered during the encoding/decoding process, while the positive element is well reconstructed. Once trained, the autoencoder can be used to explain the prediction, as described in Section 3.5. (c) represents the explanation element, where an explanation pattern is visible. As a result, the sequence $\{1 \rightarrow 0 \rightarrow 4 \rightarrow 2 \rightarrow 1 \rightarrow 5\}$ is highlighted as a representative pattern of the positive class.*

In the following, a case study is presented, when previously described methods were applied.

A patient pathway is schematically described here, along with its corresponding event log representation. An event gathers activities occurring at the same time. Medical activities found in the pathway are {A0, A1, B0, B1}, where {A0, A1} and {B0, B1} inherit from chapter A and B, respectively (like ICD-10 chapters for example). Given this knowledge regarding the hierarchy, an enriched event log is computed, by adding artificial activities representing the upper hierarchy levels. From the latter, a matrix representation is deduced, with $l = 6$ labels representing activities and hierarchy levels, and $w = 9$ time windows. White (resp. black) squares correspond to a value of 1 (resp. 0). A broader time window (e.g. weekly) would have decreased the representation precision by regrouping activities in only two columns. In that case, some activities would have occur more than once in some time windows, resulting in

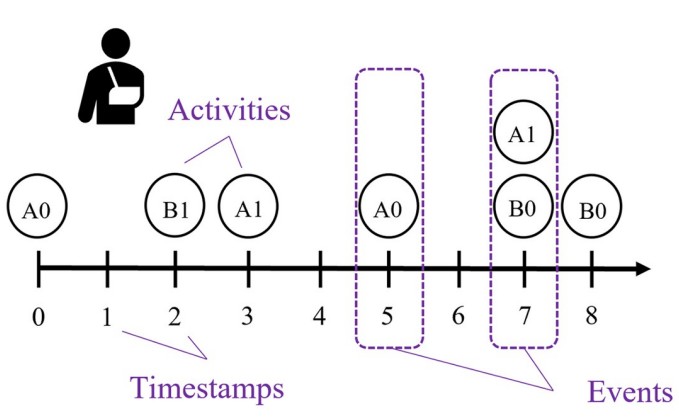

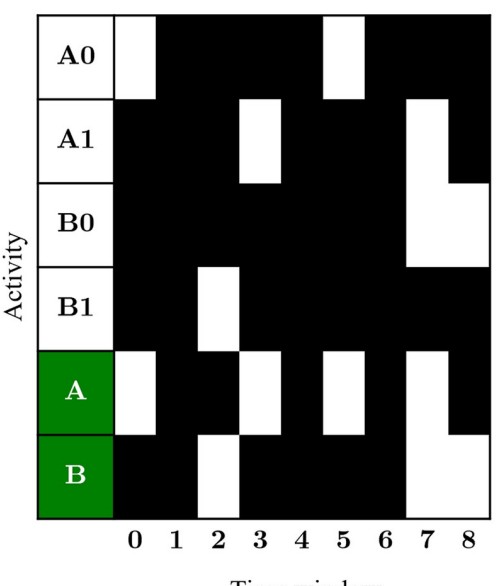

| timestamp | activity |
|-----------|----------|
| 0 | A0 |
| 2 | B1 |
| 3 | A1 |
| 5 | A0 |
| 7 | B0 |
| 7 | A1 |
| 8 | B0 |

a) Patient pathway

b) Initial event log

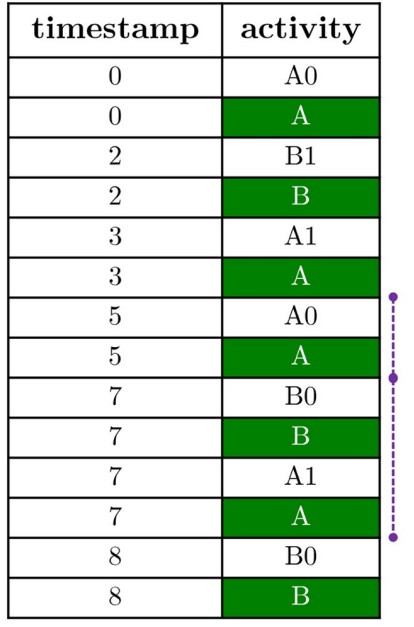

| timestamp | activity |
|-----------|----------|
| 0 | A0 |
| 0 | A |
| 2 | B1 |
| 2 | B |
| 3 | A1 |
| 3 | A |
| 5 | A0 |
| 5 | A |
| 7 | B0 |
| 7 | B |
| 7 | A1 |
| 7 | A |
| 8 | B0 |
| 8 | B |

c) Enriched event log

d) Matrix representation

**Fig 1. A complete data example: From patient pathway to its matrix representation.**

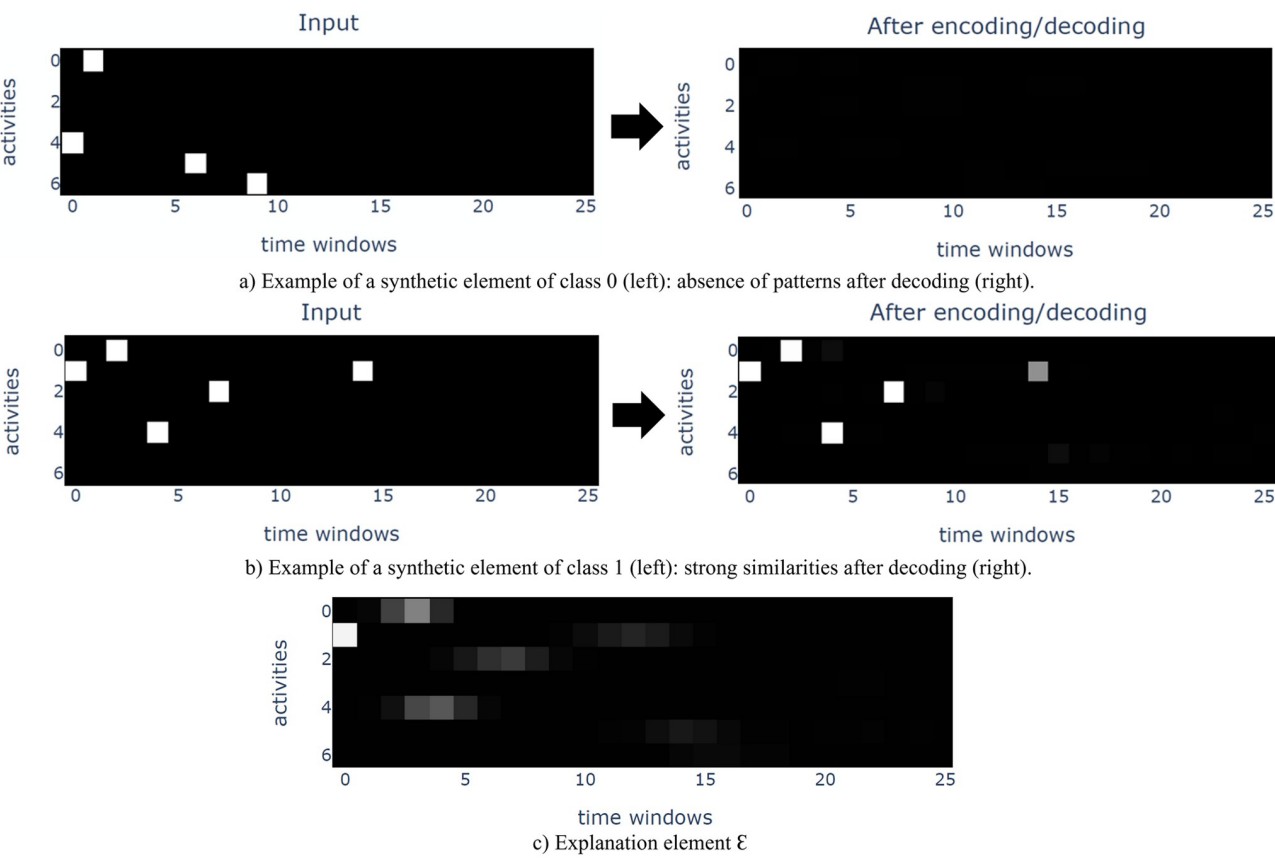

a) Example of a synthetic element of class 0 (left): absence of patterns after decoding (right).

b) Example of a synthetic element of class 1 (left): strong similarities after decoding (right).

c) Explanation element ε

**Fig 2. Examples of positive and negative synthetic elements before and after being encoded/decoded, with explanation element.**

integer values which could be rescaled between 0 and 1. Gradients of grays would have been seen in such images.

## 5 Case study

Among all deaths due to cardiovascular diseases, no less than 60% are caused by sudden cardiac death (SCD) [38]. About 3/4 of SCDs are related to ventricular tachycardia. The treatment consists in a cardiopulmonary resuscitation, combined with an electric impulse provided by an automated external defibrillator. For high-risk patients, Implantable Cardioverter-Defibrillators (ICDs) are used to prevent cardiac arrest. Once implanted, the ICD sends electric impulses to stimulate the heart in response to a potentially lethal ventricular arrhythmia. Three types of ICD exist, depending on the number of leads connecting the generator to the heart (single-lead: single chamber; two-lead: dual chamber; and three-lead: biventricular). An ICD replacement is usually necessary after several years. Possible replacement causes include a complication, a malfunction or the naturally limited durability of the device. The replacement is qualified as *short-term* if it occurs 6 to 8 years after the implantation, depending on the type of ICD.

The problem addressed here is the identification of patients with a risk of post-implantation mortality within the short-term replacement period. Moreover, the goal is to identify predictive factors in medical event logs extracted from the SNIIRAM database, considering time and hierarchy structure in medical codes, and without patient-centered information. In this

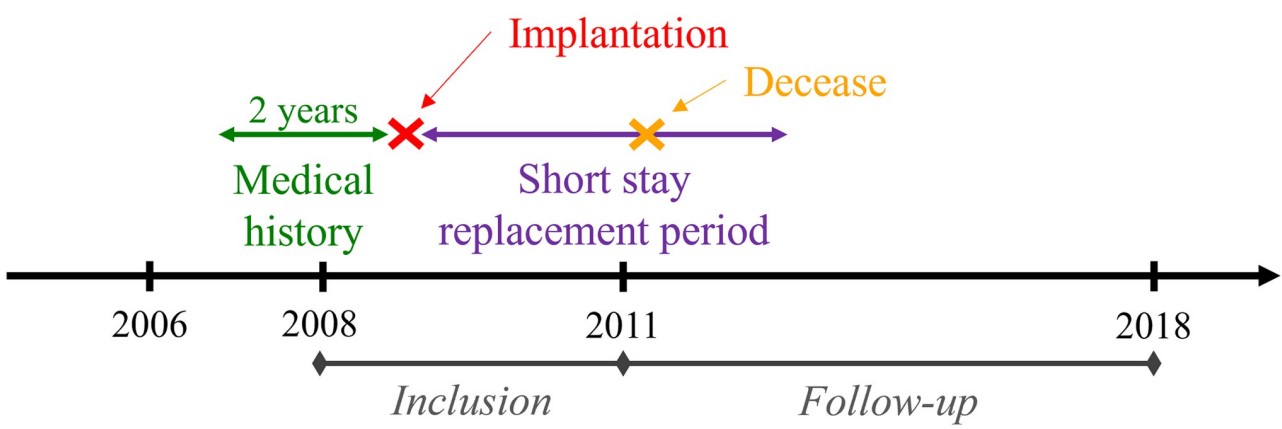

**Fig 3. Schematic representation of patients chronology.**

context, this case study serves as a proof of concept for automatic predictive factor discovery and at-risk patient identification.

## 5.1 Data preprocessing

The study included all adult patients who had an ICD implantation between 2008/01/01 and 2011/02/02 (medical procedure codes from the French Common Classification of Medical Procedures CCA: DELA004, DELF013, DELF016, DELF900, DELF014, DELF020, DELA007). Among the 19, 408 patients selected, 730 (3.8%) were excluded due to insufficient follow-up. Thus, 18, 678 patients were included in the study (5, 448, 5, 216 and 8, 014 patients having a single-, two- or three-lead ICD, respectively). The follow-up of patients was done until 2018/ 12/31 to identify potential deceases. According to medical experts' recommendations, a replacement was considered short-term if it occurs within the 8, 7, 6 years after implantation for a single-, two- or three-lead ICD, respectively. Among the population, 7, 551 patients deceased during the short-term replacement period (40.4%). For each patient, two years of medical history prior to the ICD implantation were collected. A schematic representation of patients chronology is presented in Fig 3.

Patient pathways during this period were made into an event log, which constituted the input data for the prediction. The event log regrouped the different medical events occurring over the course of the two years. Regarding hospitalizations, the reason for admittance (main diagnosis), associated diagnosis (comorbidities) and performed health care services (medical procedures and devices) were included. Other care episodes regrouped activities as consultations, biological tests and other medical procedures not performed as inpatient care. Each activity was identified by a medical code, mostly organized hierarchically: (1) ICD-10: diagnoses and comorbidities (2 levels of hierarchy); (2) CCAM: medical procedures (3 levels of hierarchy); (3) medical devices (3 levels of hierarchy); (4) biological tests (1 level in the hierarchy); (5) consultations (the code related to the type of consultation). For each medical code, upper level codes in the respecting hierarchy were added, as presented in Fig 1. Patient-centered information, such as age, gender, living localization, were not used in order to focus on the analysis of patient pathways for prediction. Based on the medical history prior to the ICD implantation, the prediction was made at the implantation discharge. The data were split in train (80%) and test (20%) sets, respecting the ratio of the two prediction classes. The training event log contained 493, 863 events and 9, 215, 335 activities. The activities and the related

**Table 2. Description of event logs, before and after filtering.**

|  | #Patients | #Activities | #Events | #Labels |
|---|---|---|---|---|
| Train | 14, 942 | 9, 215, 335 | 493, 863 | 18, 758 |
| *after filtering* | - | 8, 700, 492 | 493, 752 | 962 |
| Test | 3, 736 | 2, 344, 217 | 126, 423 | 12, 282 |
| *after filtering* | - | 2, 209, 349 | 126, 396 | 962 |

hierarchy levels were filtered to discard infrequent elements (threshold: 500 occurrences). The filtering evacuated non-representative codes, while keeping widely represented ones and higher levels in the hierarchy. The number of different labels decreased from 18758 to 962, while keeping more than 99, 9% of events and 94, 4% activities for the training event log. The test event log was filtered by keeping the selected labels. Description of the event logs are presented in Table 2.

## 5.2 Methods

In order to identify patients with a risk of post-implantation mortality within the short-term replacement period, several predictive methods were used, divided in three categories: machine learning (ML), deep learning (DL) and autoencoders (AE).

**5.2.1 Machine learning.** Decision Tree (DT), Random Forest (RF), Logistic Regression (LR) and LightGBM [39] (LGBM) were deployed in this study. As these methods take patient-centered vectors as input, two methods were used to transform event logs into adequate features. The first one is the use of a list of features (LOF). For each one of the $l = 962$ selected labels, the total number of appearance in the event log was computed and added as a feature. In order to provide a temporal dynamic to ML methods, the second one uses four time windows (TW), with a list of features computed for each time window, resulting in $4 \times l = 3, 848$ features. Hyperparameters of each ML method were optimized using 50 iterations of Optuna [40], evaluating the average of the area under the receiver operating characteristic curve (AUC-ROC) on a 5-folds validation of the training data.

**5.2.2 Deep learning.** 5 deep learning architectures were constructed and trained in order to predict the post-implantation mortality. Each architecture is constituted of a succession of three layers, and uses batch normalization to speed up convergence and dropout to limit overfitting. The first one, referred to as "Dense", uses a first dense layer to encode activities, a flatten and dense layer to merge time-windows, and a final dense output for prediction. The four other architectures replace the middle flattening and dense layer by a recurrent one. The tested recurrent layers were Long-Short Term Memory (LSTM), Gated Recurrent Units (GRU), and the bidirectional correspondence (BiLSTM and BiGRU, respectively). Hyperbolic tangent as activation functions were used, except for the output (sigmoid). The dimension of the layers was set to 32, and the loss function used was the binary cross entropy.

**5.2.3 Autoencoders.** In order to test the previously defined objective functions $\mathcal{J}_\theta^F$ and $\mathcal{J}_\theta^I$, three autoencoding architectures were constructed. The first one (Regular AE), uses an encoder to project input data into a lower dimension space, and a decoder to reconstruct data from this representation back to the initial space. The second one is a Denoising AutoEncoder (DAE), which has a similar architecture but add some noise to input data during the training. The noise is defined as choosing random elements $x_{i,j} \in x$ and replacing them by their inverse $1 - x_{i,j}$. The ratio of elements taken randomly to be inversed is set to 0.1%. The last autoencoding method tested was a Variational AutoEncoder (VAE) [41]. A Variational AutoEncoder (VAE) is an autoencoder where the learnt variables are parameters of a distribution. The

training process consists of maximizing the evidence lower bound (ELBO). In practice, the single sample estimate $\log p(x|z) + \log p(z) - \log q(z|x)$ with $z$ sampled from the inference network is optimized. To adapt the *ELBO* with the previously defined objective functions $\mathcal{J}_\theta^F$ and $\mathcal{J}_\theta^I$, the term $\log p(x|z)$ which measures in practice the reconstruction error, is replaced by $\mathcal{J}_\theta^F$ or $\mathcal{J}_\theta^I$. The latent dimension $d_{latent}$ is set to 8. As the VAE learns a probabilistic representation of data, a set of means and variances are outputted by the encoder. Thus, $d_{latent}^* = 2 \times d_{latent}$ for VAE, $d_{latent}^* = d_{latent}$ otherwise. Details about these architectures are provided in Fig 4.

For DL and AE methods, the event logs were transformed into three dimensional `NumPy` arrays [42], following the modeling described in Section 4. For each patient, its history was represented as a matrix of $l \times w$ with $l = 962$ the number of retained labels and $w = 24$ the number of time windows. If $n$ denotes the number of patients in the considered event log, the final

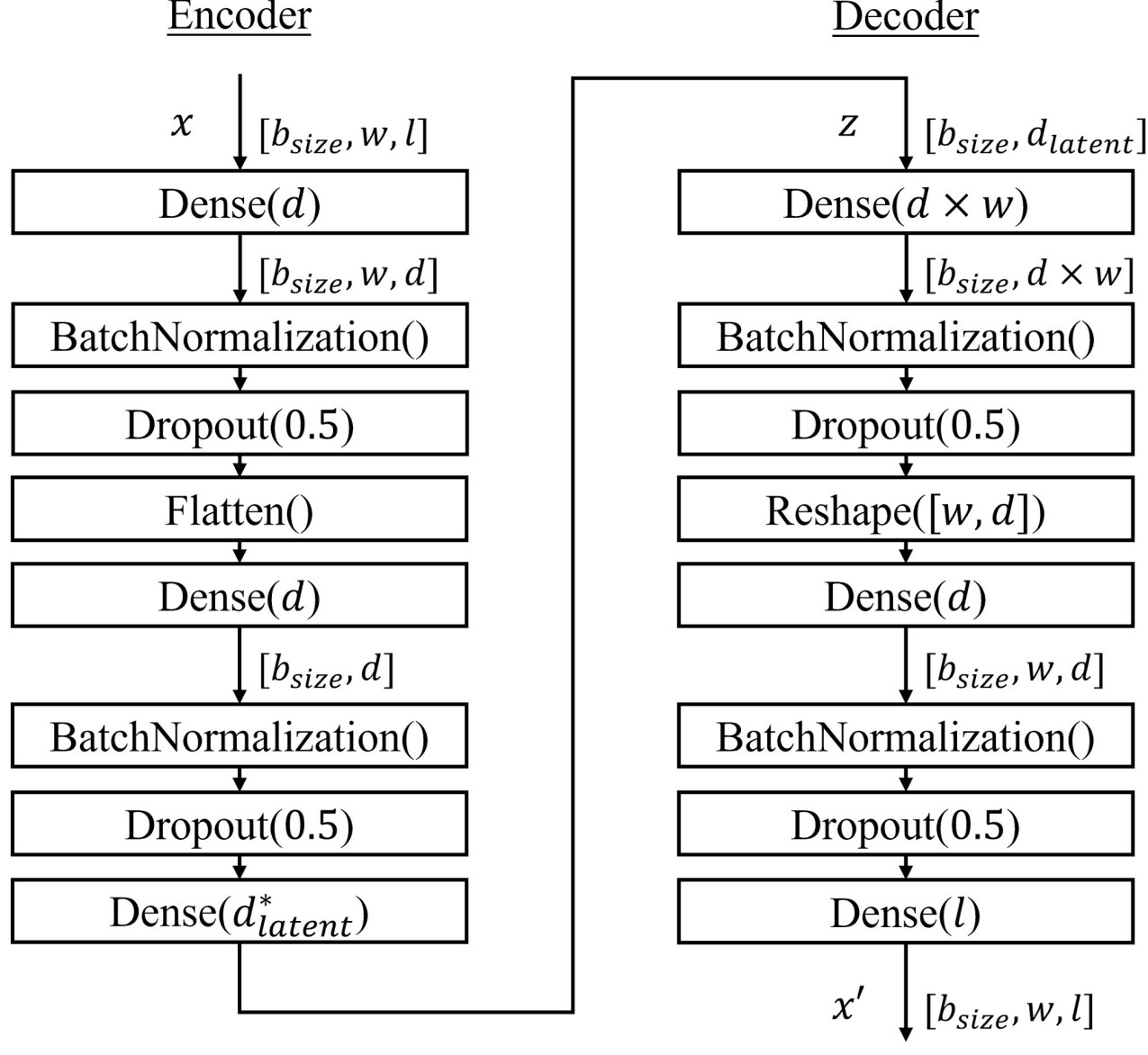

**Fig 4. Schematic representation autoencoding architectures.**

shape of data is then ($n \times w \times l$). In order to measure the predictive performances, three quantitative measures were evaluated on the test set: the area under the receiver operating characteristic curve (AUC-ROC), the area under the precision-recall curve (AUC-PR), and the Matthews correlation coefficient (MCC). For the last measure, a threshold was automatically computed for autoencoders by minimizing the gini impurity for the positive and negative output distributions. The output is defined by the decoded information for $\mathcal{J}_\theta^F$ and by the reconstruction error for $\mathcal{J}_\theta^I$. Each model was trained using three-dimensional data. Adam [43] algorithm was used for optimization (learning rate = $1e^{-4}$), with a maximum of 5, 000 epochs and an early stopping strategy of 25 epochs patience. The batch size $b_{size}$ was set to 64. The dimension $d$ of the layers was set to 32.

**5.2.4 Computational details.** The experiments were coded in Python 3.7 and performed on an Intel Core i5 processor (2.6 GHz), 16 GB RAM, and Windows 10 OS. DT, RF and LR were implemented using `scikit-learn` [44]. Deep learning and autoencoder models were implemented using `tensorflow` [45].

## 5.3 Quantitative results

Quantitative results are presented in Fig 5, where the proposed methods have been compared to other explainable and widely used methods (a), as well as to black-box methods used more recently in the literature (b). All the methods show encouraging performances, illustrating that non-clinical claims data are interesting resources for predictive studies, even without patient characteristics. Results show that proposed objective function $\mathcal{J}_\theta^I$ outperforms $\mathcal{J}_\theta^F$. In other words, filtering only positive examples during training appears to be insufficient, and inverting the data for negative examples seems to be useful in order to improve performances. In particular, $\mathcal{J}_\theta^I$ with a VAE architecture provides better results in comparison with explainable methods such as DT and LR (with both LOF and TW features), but also in comparison with RF and the deep learning architecture with dense layers. The best results in AUC-ROC (0.75) and AUC-PR (0.68) are obtained by deep learning methods with recurrent layers (BiGRU and LSTM). Nevertheless, the proposed method $\mathcal{J}_\theta^I$ with VAE appears as a close second, with an AUC-ROC of 0.74 and an AUC-PR of 0.67. LGBM obtains similar results regarding these two scores. Regarding MCC, $\mathcal{J}_\theta^I$ with VAE provides the best results (MCC of 0.37). These results show the competitiveness of the method with black-box models, the slight performance loss being compensated by the gain in explainability.

## 5.4 Explainability of prediction

**5.4.1 Predictive factors.** The results presented in the following focus on the explainability of the autoencoder method with objective function $\mathcal{J}_\theta^I$ on a VAE architecture, and are summarized in Fig 6. As presented in Section 3, a global explanation of the prediction can be produced from training data by computing the explanation element $\mathcal{E}$. By encoding and decoding both positive and negative patients, and evaluating the difference, characteristic patterns of the positive class appears. As the representation of one pathway is a 2-dimensional array, Fig 6(a) shows $\mathcal{E}$ represented as an image. Here, two patterns emerge from the explanation element: (1) continuous horizontal lines over the two years; and (2) particular events occurring in the last time window before implantation. This implies that both some long-term and recurrent medical events, but also punctual and last-month events, have an influence on the prediction target. To verify these assumptions, relative risks are computed among the entire population $X$ of size $n$ (train and test), by following the method described thereafter.

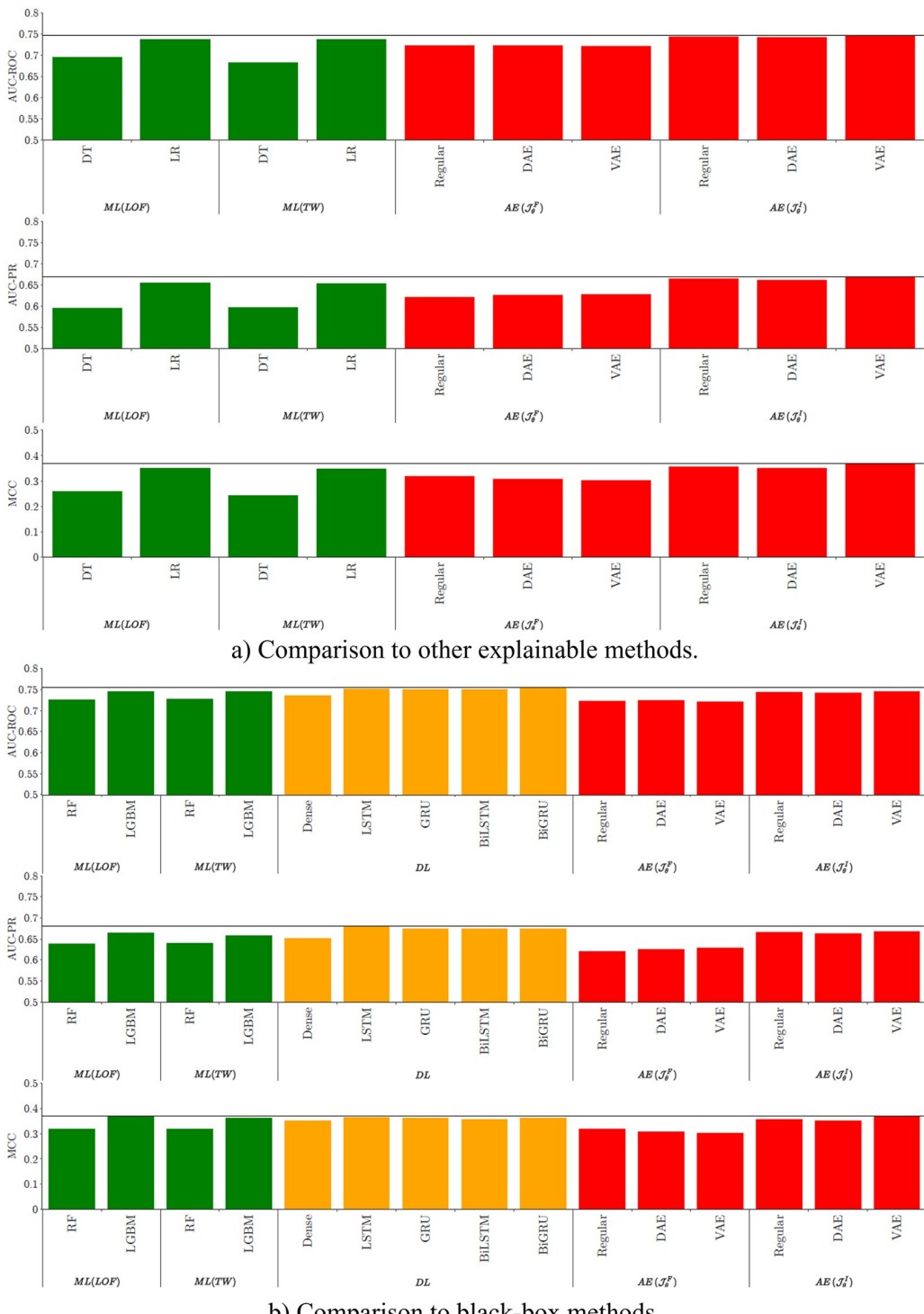

a) Comparison to other explainable methods.

b) Comparison to black-box methods.

**Fig 5. Quantitative results of the explainable (a) and black-box (b) methods regarding AUC-ROC, AUC-PR and MCC, evaluated on test data.** The black lines represent the best scores obtained for each comparison.

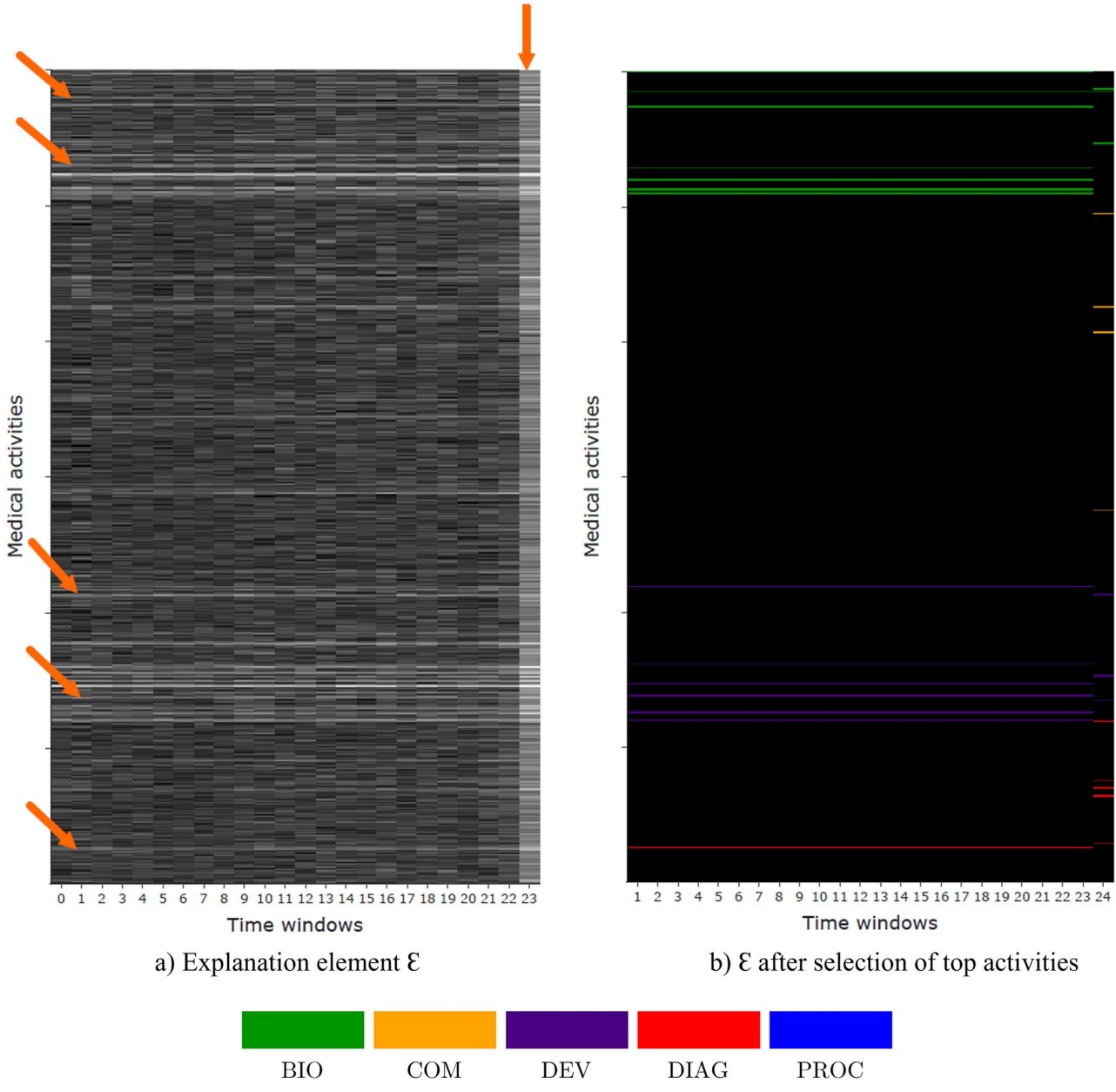

a) Explanation element $\mathcal{E}$  b) $\mathcal{E}$ after selection of top activities

| BIO | COM | DEV | DIAG | PROC |

**Fig 6.** Explanation using $\mathcal{E}$ (left, $\mathcal{J}_\theta^l$ with a VAE architecture). Here, continuous horizontal lines over the two years, as well as particular events occurring in the last time window before implantation are highlighted (arrows). These patterns are clearly identified after selecting only top decoded activities (right), where each colour corresponds to a particular type of medical activity.

**5.4.2 Specific validation methodology.** Firstly regarding frequent events, top-100 activities $i$ with higher $\sum_{j=1..w-1}\mathcal{E}_{i,j}$ are selected (corresponding to activities with higher values strictly before the last time window). Then, for each of the most decoded activities $i$, patients are split in two groups: those with reoccurring activities (patients with activity in the pathway superior to a threshold defined as the mean for the considered activity $i$, i.e. $x \in X$ such as $\sum_{j=1..w-1}x_{i,j} \geqslant \frac{1}{n}\sum_{x \in X}\sum_{j=1..w-1}x_{i,j}$) versus others. Relative risks are finally computed regarding these groups. The same process is applied for the last time window before implantation. For

both frequent and last time window's, activities with the most significant relative risks are selected as explaining factors. Therefore, selected activities are filtered on $\mathcal{E}$ and highlighted in Fig 6(b), where the colors corresponds to the type of activity (BIO: biology, COM: comorbidity, DEV: medical device, or DIAG: diagnosis). The top-15 activities summarizing $\mathcal{E}$ in explainable results is presented in Fig 7. The latter provides a minimal representation of the global explanation deduced from the model, within an interpretable image of the pathways. Relative risks with a 95% confidence interval for these frequent and last time window activities are presented in Fig 8(a) and 8(b), respectively.

**5.4.3 Relative risks analysis.** Results show that the 15 highlighted risk factors appears to be significant regarding the considered outcome, with relative risks between 1.2 and 1.8. Regarding recurrent events during the past two years (a), results shows that recurrent biological tests and related medical devices have a significant impact on the prediction target. Moreover, devices related to heart diseases (leads for pacemakers and ICDs) and recurrent hospitalizations for circulatory diseases appears as related risk factors identified by the proposed method. Regarding the last time window (b), hospitalizations for chronic obstructive pulmonary diseases are highlighted, as well as for circulatory diseases (atrial fibrillation and congestive heart failure). Comorbidities of the patients which have been noted during event of the last time window are also highlighted, such as acute renal failure, acute bronchitis, heart failure or type 2 diabetes. The use of screws for osteosynthesis implants, and of human origin implants involving several anatomical devices also point out major surgery during the last month before the implantation as high risk factors. One can note that various levels of the hierarchy of codes are selected as predictive factors. Also, no particular medical procedure is part of the top activities.

As a result, the presented case study demonstrates the ability of autoencoders to identify predictive factors from event logs extracted from a claim database. Starting from anonimyzed event logs without patient characteristics, the presented method allows for a representation of data which is able to model the hierarchical code structures and temporal information. Prediction performances are competitive with black-box models, while being explainable, as illustrated by the identification of infrequent short-term factors and frequent long-term ones. A verification of the implication of such factors with the observed outcome was also presented using relative risks. These results validate the ability of the method for the discovery of predictive factors from raw event logs.

# 6 Conclusion and future work

In this paper, the use of autoencoders to explain predictive factors in patient pathways has been presented. The related formalization, including objective functions and methodologies to both predict and explain have been proposed. An adapted method to model and transform complex medical event logs for that purpose has been presented, with a focus on time and hierarchy modeling. A thorough case study on real data extracted from the SNIIRAM database was also conducted. The short-term mortality after the implementation of an ICD has been predicted, reaching performances which are in accordance to the literature. Moreover, the ability of the method to provide explainable predictions has been illustrated, and the predictive factors have been validated using relative risks.

Some limitations and future perspectives are discussed in the following. Qualitative comparison of explainable factors extracted by the proposed method with factors extracted by other explainable methods have not been conducted. Such a comparison study, including also post-hoc explainable methods, would have been interesting to conduct and will be part of future work. Two different strategies ("filter" and "inverse") have been proposed to construct

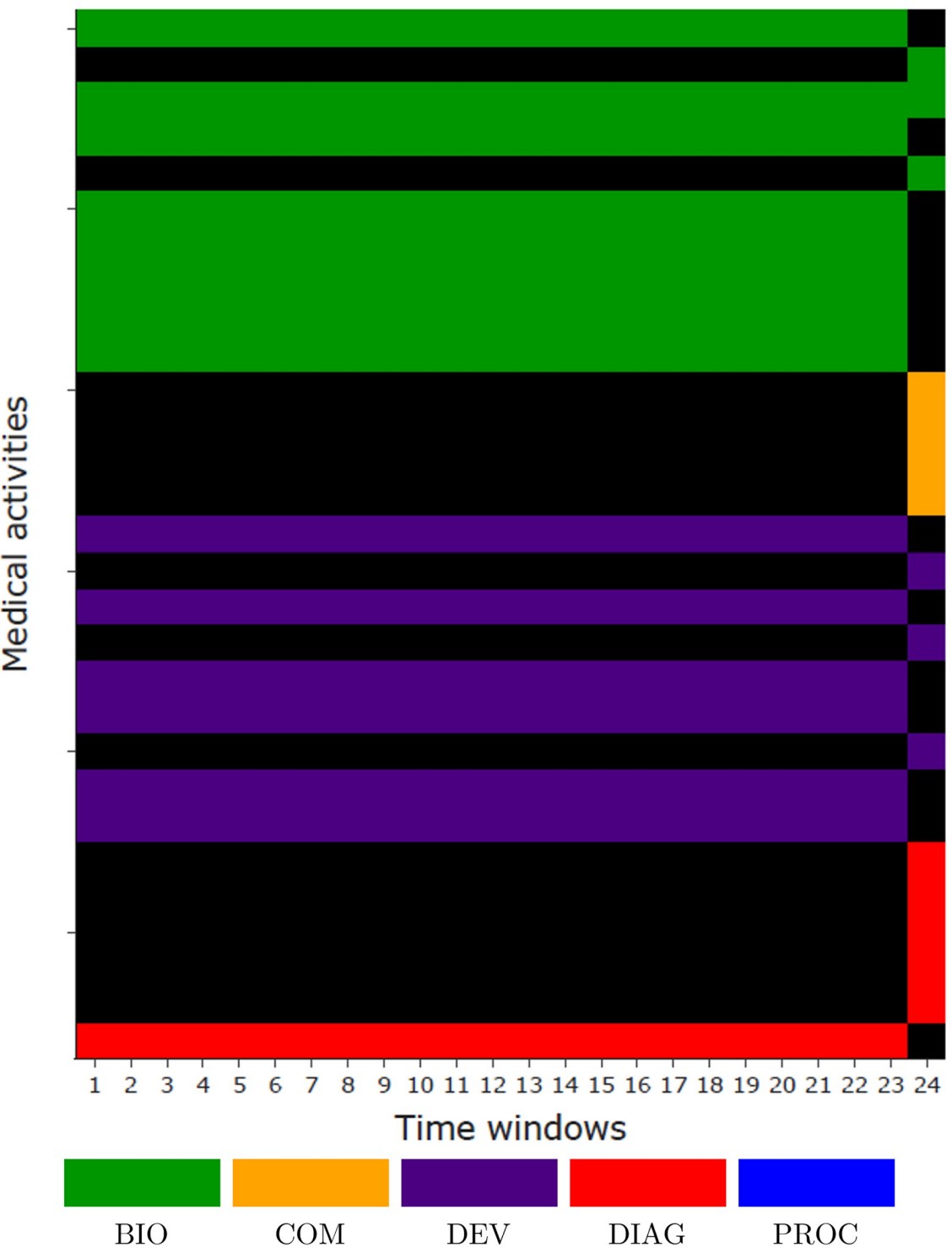

**Fig 7. Minimal representation of $\mathcal{E}$.**

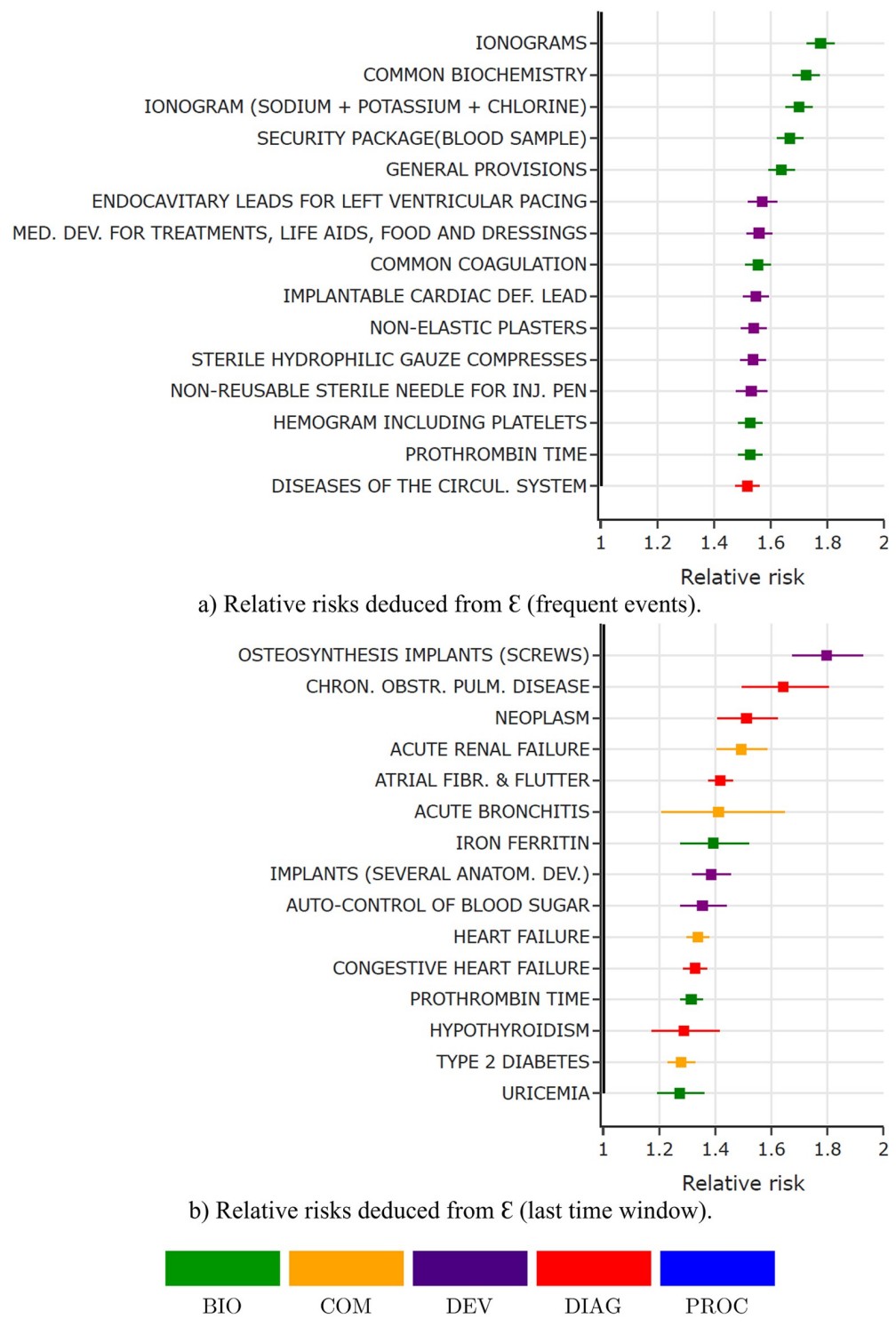

a) Relative risks deduced from $\varepsilon$ (frequent events).

b) Relative risks deduced from $\varepsilon$ (last time window).

BIO   COM   DEV   DIAG   PROC

**Fig 8. Validation of explanation using relative risks ($\mathcal{J}_\theta^I$ with a VAE architecture).**

the loss function and the corresponding output, that derives both the classification process and the explanation. Testing other strategies is also part of future research. The size of the time window is a parameter that needs to be defined by the user, derived in practice by medical knowledge of the pathology. Future improvements of the method would focus on the creation of a methodology to choose the size of the time window directly from the data, and validate such methodology regarding both predictive performances and explanations. Moreover, improving the method by using time windows that may be different for each activity is a leek for future work. Even if this work shows the power of claims for predictive purposes, performances may be not sufficient for practical applications. In order to improve performances, the use of different architectures for the encoder and the decoder should be considered in future research. Recurrent layers for the encoder and the decoder were implemented and tested, achieving better predictive performances but without revealing particular predictive factors. More investigations on the stability of the proposed formalization regarding explainability are part of the future work. While this paper focuses on patient pathways, the method can also be extended in order to include patient characteristics. This can lead to better results for pathologies where such characteristic could impact the predicted outcome, but also lead to different pathways and thus different predictive factors to identify depending on the sub-population. Other medical information (e.g. image, vital signs, free text, etc.) could also be processed using deep learning architectures. The representation of such data in order to explain such results should be considered in future research. Nevertheless, the main perspectives will concern the deployment of this methodology to other medical case studies. This will properly evaluate the benefits offered by such methodologies for the health system, in terms of data-driven targeted prevention policies. The explainability in this context could produce knowledge directly from patient pathway data. This can be beneficial to generalize guidelines from the identified predictive factors, in order to perform early at-risk patient detection. In a more exploratory context, weak signals could be detected by following this approach, leading to the formulation of hypotheses to test and loop on the data using more humanly comprehensive indicators, like descriptive statistics and relative risks. Designing such an automated multi-pathology framework for the detection of predictive factors is a promising future work.

## Supporting information

**S1 Appendix. Supplementary materials.** The supplementary materials include details on the conducted experiments (hyperparameters, deep learning architectures), as well as all the quantitative and qualitative results obtained.
(PDF)

## Acknowledgments

The authors wish to thank Claire Leboucher for her help with data management.

## Author Contributions

**Conceptualization:** Hugo De Oliveira, Augusto Vincent, Xie Xiaolan.

**Data curation:** Hugo De Oliveira.

**Formal analysis:** Hugo De Oliveira.

**Methodology:** Hugo De Oliveira.

**Project administration:** Prodel Martin.

**Software:** Hugo De Oliveira.

**Supervision:** Prodel Martin, Lamarsalle Ludovic, Augusto Vincent, Xie Xiaolan.

**Visualization:** Hugo De Oliveira.

**Writing – original draft:** Hugo De Oliveira.

**Writing – review & editing:** Hugo De Oliveira, Prodel Martin, Augusto Vincent.

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
