## [Decision Letter · Decision Letter 0]

6 Jul 2022

PONE-D-22-13428Explaining predictive factors in patient pathways using autoencodersPLOS ONE

Dear Dr. De Oliveira,

Thank you for submitting your manuscript to PLOS ONE. After careful consideration, we feel that it has merit but does not fully meet PLOS ONE’s publication criteria as it currently stands. Therefore, we invite you to submit a revised version of the manuscript that addresses the points raised during the review process.

ACADEMIC EDITOR: Please consider and address the comments from the reviewers very carefully. In addition to the technical content, language revision is also required.==============================

We look forward to receiving your revised manuscript.

Kind regards,

Nattapol Aunsri, Ph.D.

Academic Editor

PLOS ONE

Journal Requirements:

Reviewers' comments:

Reviewer's Responses to Questions

**Comments to the Author**

1. Is the manuscript technically sound, and do the data support the conclusions?

Reviewer #1: Yes

Reviewer #2: Yes

2. Has the statistical analysis been performed appropriately and rigorously? 

Reviewer #1: Yes

Reviewer #2: Yes

3. Have the authors made all data underlying the findings in their manuscript fully available?

Reviewer #1: No

Reviewer #2: Yes

4. Is the manuscript presented in an intelligible fashion and written in standard English?

Reviewer #1: Yes

Reviewer #2: Yes

5. Review Comments to the Author

Reviewer #1: In this article titled “Explaining predictive factors in patient pathways using autoencoders”, the authors investigated the performance of autoencoders to explain predictive factors in patient pathways. To develop the above, the authors develop a method to transform and represent complex medical event logs. The developed autoencoders have been compared with other deep learning and machine learning methods by testing on retrospective data from the SNIIRAM database, and demonstrates competitive prediction performances. The use of autoencoders for explaining predictive factors in patient pathways is an interesting application. However, the document can be improved and made clearer to the audience. The grammar of the manuscript needs to be improved and the methodology-result section needs some rearrangement. In addition, discussion on certain aspects of the results is lacking.

General issues/questions

1) Section 3 – line 202 (below eqn 8): The meaning of the sentence “ The explanation element…that the any input elements…” is not clear due to apparent mistake in sentence structure. Consider rewriting.

2) Section 5.2.1 – line 322: The acronym AUC-ROC was not defined, but was only defined in a later Section 5.2.3 (line 360).

3) Section 5.2.2: The authors state the selection of activation functions, layer dimension and loss function. Any particular reason why these hyperparameters were chosen? Was there any analysis/ hyperparameter tuning being made to optimise these hyperparameters, particularly because they are the same for all five DL models?

4) Section 5.2.3 – line 348: acronym ELBO is not defined

5) Section 5.2.3 - line 365: “DT, RF and LR…. [43]. Deep learning… [44]”. These sentences are general statements (not specific to the autoencoder section). It is suggested that a separate section (5.2.4) be added to include these statements as to provide clarity and avoid confusion to the readers. The last sentence of Section 5.2.3 “The experiments…. Windows 10 OS” could also be shifted to this newly added section.

6) Section 5.3 and Fig 5: The authors present quantitative results, including values for the AUC-ROC, AUC-PR, and MCC. The reviewer suggests that the authors give a simple explanation of these results, particularly what these values represent in relation to the case study presented? Also, any particular reason of the discrepancy in results between AE(J_I) and AE(J_F)? This should also be included in the discussion.

7) Section 5.4: The authors use “(a)” and “(b)” to reference to Fig 6. It is suggested that the authors use “Fig 6 (a)” and “Fig 6 (b)” as to improve clarity and avoid confusion. furthermore, “(c)” is mentioned in line 414, however such figure does not exist in Fig. 6. The reviewer believes this refers to Fig 7 instead, please amend.

8) Section 5.4 – line 416,417: The authors mention that “the relative risks with a 95% confidence interval….are presented in (a) and (b), respectively”. Again, this should be in reference to Fig 8 but was not mentioned. Please amend.

9) Section 5.4: the authors presented a method of validating the assumptions made from the results of Fig 6a by computing the relative risks. The following text describing this methodology should be exclusive to the methods section, rather than the results/discussion.

10) Conclusion – line 456: The authors mentioned that better strategies can be used to improve prediction performance and explainability. Could the authors provide some possible examples/potential methods to achieve this? If not, this sentence remains highly speculative.

11) This study focuses exclusively on patient pathways to explain predictive factors, while omitting patient characteristics such as sex, age, and race. Would the inclusion of these characteristics in the study improve performance and/or uncover hidden patterns? How would the omission of these variables affect potential clinical applications? As the mortality or other patient outcomes could be affected by such factors , ie there is an inherent predisposition of the patient due to these factors.

Minor issues:

1) Section 3 – line 157: “…input data x and return a lower…”. Should be “returns”

2) Section 4 – line 221: “..the considered case study”. Missing fullstop.

3) Section 4 – line 225: “..all level of…”. Should be “all levels of”

4) Section 5.2.1 – line 322: “…, evaluation the mean…” should be “…and evaluating the mean…”

5) Section 5.2.2 – line 330: “The 4 other architecture replace…” should be “The 4 other architectures replace…”. Also, suggested to use spelling for numbers that are less than 10, ie “four” instead of “4”.

6) Section 5.2.3 – line 346: “The training process consists in…”. Should it be “consists of”?

7) Section 5.2.3 – line 354: “For DL ans AE…”. Typo of the word “and”

8) Section 5.3 – line 373 “…, were the proposed methods…” should be “…, where the proposed methods…”.’

9) Section 5.4 – line 431: “…that various level…” should be “…that various levels”

10) Section 5.4 – line 393: “…train data by computing…” should be “…training data by computing…”

11) Conclusion – line 460: “..of the time widow”. Typo of the word “window”

Reviewer #2: Explaining predictive factors in patient pathways using autoencoders

PONE-D-22-13428

The abstract:

After reading the whole paper, I found that the abstract lacks the jism of the work. The work done is not reflected in the abstract. I would advise to review same. I understand that there is a limitation on the number of words for the abstract- However, it lacks the main aspect of the work.

Introduction:

Provided the research gap and the challenges that need to be addressed.

The main contribution is to explain the causal factors and to devise a framework to model the system

Introduction is explicit and well- written

Literature review

Right information to understand the topic

Section 3:

Section 3 focuses on autoencoders- However, this section seems to be disjoint over the whole paper. The explanations with the formula are good. However, it would be better if this section was linked with the work conducted in this paper and not just a preliminary where a reader does not have the interest why this is being discussed here.

Section 5:

Presents the case study. From what is reported is that a satisfactory amount of data was captured from 18, 678 patients

The authors have used machine learning (ML), deep learning (DL) and autoencoders (AE) to select and represent data. There are not enough details regarding deep learning.

Explainability of the parameters provided- However, the authors have not compared it with other author’s works. I understand that it is different, since it shows the explainability part- However, there is a need to provide a discussion in relation with other work.

One of the contributions of this paper is to validate the predictive factors extracted through relative risks, widely used in bio-statistical analysis. Some missing discussions in relation to the results and other people’s work.

Otherwise, it is a good piece of work.

6. PLOS authors have the option to publish the peer review history of their article (what does this mean?). If published, this will include your full peer review and any attached files.

Reviewer #1: No

Reviewer #2: No

---

## [Author Response · Author response to Decision Letter 0]

28 Aug 2022

Dear reviewers, 

Thank you very much for the interesting comments and useful suggestions provided regarding the manuscript entitled “Explaining predictive factors in patient pathways using autoencoders”.

By following your recommendations, a new version of the manuscript has been submitted for publication. Majority of your comments have been addressed and your suggestions have been included in this new version. In order to facilitate the review, a point-by-point answer to your comments is also attached to the resubmission.

Therefore, we truly hope that changes applied will satisfy all the requirements, making our paper suitable for publication in PLOS ONE.

Sincerely,

Hugo De Oliveira, on behalf of Martin Prodel, Ludovic Lamarsalle, Vincent Augusto and Xiaolan Xie.

---

## [Decision Letter · Decision Letter 1]

21 Oct 2022

Explaining predictive factors in patient pathways using autoencoders

PONE-D-22-13428R1

Dear Dr. De Oliveira,

We’re pleased to inform you that your manuscript has been judged scientifically suitable for publication and will be formally accepted for publication once it meets all outstanding technical requirements.

Kind regards,

Nattapol Aunsri, Ph.D.

Academic Editor

PLOS ONE

Additional Editor Comments : -

Reviewers' comments:

Reviewer's Responses to Questions

**Comments to the Author**

1. If the authors have adequately addressed your comments raised in a previous round of review and you feel that this manuscript is now acceptable for publication, you may indicate that here to bypass the “Comments to the Author” section, enter your conflict of interest statement in the “Confidential to Editor” section, and submit your "Accept" recommendation.

Reviewer #1: All comments have been addressed

2. Is the manuscript technically sound, and do the data support the conclusions?

Reviewer #1: Yes

3. Has the statistical analysis been performed appropriately and rigorously? 

Reviewer #1: Yes

4. Have the authors made all data underlying the findings in their manuscript fully available?

Reviewer #1: Yes

5. Is the manuscript presented in an intelligible fashion and written in standard English?

Reviewer #1: Yes

6. Review Comments to the Author

Reviewer #1: The authors carefully considered and answered the reviewers' comments and questions. Thank you for the excellent job! The revised paper is a significantly improved version of the original manuscript, it is worth publishing.

7. PLOS authors have the option to publish the peer review history of their article (what does this mean?). If published, this will include your full peer review and any attached files.

Reviewer #1: No
